# Efficacy of adjuvant chemotherapy with S-1 in stage II oral squamous cell carcinoma patients: A comparative study using the propensity score matching method

**Ryoji Yoshida**[1]*, **Masashi Nagata**[1☉], **Akiyuki Hirosue**[1‡], **Kenta Kawahara**[1‡], **Masafumi Nakamoto**[1‡], **Masatoshi Hirayama**[1‡], **Nozomu Takahashi**[1‡], **Yuichiro Matsuoka**[1‡], **Junki Sakata**[1‡], **Hikaru Nakashima**[1‡], **Hidetaka Arita**[1‡], **Akimitsu Hiraki**[2‡], **Masanori Shinohara**[3‡], **Ken Kikuchi**[4‡], **Hideki Nakayama**[1‡]

**1** Department of Oral and Maxillofacial Surgery, Faculty of Life Sciences, Kumamoto University, Kumamoto, Japan, **2** Department of Oral and Maxillofacial Surgery, Section of Oral Oncology, Fukuoka Dental College, Fukuoka, Japan, **3** Itoh Dent-Maxillofacial Hospital, Kumamoto, Japan, **4** Sakurajyuji Hospital, Kumamoto, Japan

☉ These authors contributed equally to this work.
‡ These authors also contributed equally to this work.
* ryoshida@kumamoto-u.ac.jp

**Data Availability Statement:** All relevant data are within the paper.

## Abstract

It has been reported that 20% of early-stage oral squamous cell carcinoma (OSCC) patients treated with surgery alone (SA) may exhibit postoperative relapse within 2–3 years and have poor prognoses. We aimed to determine the safety of S-1 adjuvant chemotherapy and the potential differences in the disease-free survival (DFS) between patients with T2N0 (stage II) OSCC treated with S-1 adjuvant therapy (S-1) and those treated with SA. This single-center retrospective cohort study was conducted at Kumamoto University, between April 2004 and March 2012, and included 95 patients with stage II OSCC. The overall cohort (OC), and propensity score-matched cohort (PSMC) were analyzed. In the OC, 71 and 24 patients received SA and S-1, respectively. The time to relapse (TTR), DFS, and overall survival were better in the S-1 group, but the difference was not significant. In the PSMC, 20 patients each received SA and S-1. The TTR was significantly lower in the S-1 group than in the SA group, while the DFS was significantly improved in the former. S-1 adjuvant chemotherapy may be more effective than SA in early-stage OSCC.

## Introduction

Oral cancer, and predominantly oral squamous cell carcinoma (OSCC), is a major cause of morbidity and mortality worldwide. The survival rate of these patients has not improved, despite advances and innovations in the diagnostic techniques and treatments used [1]. Locally advanced OSCC is generally associated with particularly poor prognoses owing to the difficulty in controlling it with surgery and adjuvant radiotherapy or concurrent chemoradiotherapy [2, 3]. However, even in case of early-stages disease (T1-2N0), which can be cured by therapy, more than 80% of the cases being subjected to curative surgery may exhibit postoperative

**Funding:** This work was supported by Grant-in-Aid for Scientific Research C (18K09771).

**Competing interests:** The authors have declared that no competing interests exist.

relapse within the first 2 to 3 years [4, 5]. Therefore, it is essential to control local recurrence and/or regional lymph node metastasis to improve the patients' prognoses.

S-1, a novel oral fluoropyrimidine preparation (Taiho Pharmaceutical, Tokyo, Japan), is designed to improve the antitumor activity of 5-FU, while also reducing gastrointestinal toxicity. S-1 contains tegafur (a prodrug of 5-FU), gimeracil (inhibits the 5-FU degeneration enzyme, dihydropyrimidine dehydrogenase), and oteracil (reduces the gastrointestinal toxicity of 5-FU) [6–8]. In patients with various cancers, including those of the head and neck, S-1 administered alone or in combination with other chemotherapeutic agents has been shown to improve the outcomes [9–12].

Recently, the Adjuvant Chemotherapy with S-1 after Curative Treatment in Patients with Head and Neck Cancer (ACTS-HNC) study, which enrolled patients with advanced head and neck squamous cell carcinoma (HNSCC), reported significantly better overall survival (OS) in the S-1 group than in the control group [13]. Furthermore, reports have confirmed the efficacy of S-1 after curative surgery in gastric and pancreatic cancer [10, 14, 15]. These results encouraged us to investigate whether S-1 could be considered as a treatment option after curative surgery in patients with OSCC, even in early-stage disease. The primary aim of this study was to evaluate the efficacy and safety of S-1 compared with surgery alone (SA) in patients with stage II OSCC.

## Material and methods

### Study population

The study population comprised patients with cT2N0 (stage II) OSCC. They were diagnosed based on the histological and radiological findings, including computed tomography (CT), magnetic resonance imaging, ultrasonography, and positron emission tomography-computed tomography (PET-CT) findings. All tumors were staged according to the TNM classification of the American Joint Committee on Cancer, 7th edition [16], and the degree of differentiation was determined according to the classification of the World Health Organization [17]. Histopathological tumor invasion phenotypes were categorized with respect to the mode of invasion [18]. In our department, elective neck dissection is only performed for cases of N0 oral cancer for the purpose of reconstruction, and a "wait-and-see policy" has been adopted. Therefore, all patients enrolled in the present study only underwent resection of the primary tumor. After completion of curative surgery, we confirmed whether the patients met the eligibility criteria. Those treated with SA and S-1 at the Department of Oral and Maxillofacial Surgery, Kumamoto University Hospital were enrolled from April 2004, with observation continuing until March 2012. This study was conducted with the approval of the Ethics Committee of Kumamoto University (approval number, 747), in accordance with the guidelines for Good Clinical Practice and the Declaration of Helsinki. All patients provided written informed consent before enrollment in the study.

### Eligibility criteria

To further improve the outcomes in patients with early-stage OSCC, we began administering adjuvant chemotherapy with S-1 from 2004. The eligibility criteria for adjuvant chemotherapy with S-1 for patients with cT2N0 oral carcinoma were as follows: (1) curative surgery only for the primary tumor, (2) histologically verified SCC of the oral cavity, (3) no residual tumor (primary lesion) confirmed on diagnostic imaging or biopsy, (4) performance status of 0–1 and normal hematologic parameters (white blood cell count $\geq$ 3500/mm$^3$, hemoglobin level $\geq$ 9.0 g/dL, and platelet count $\geq$ 100,000/mm$^3$), liver function (total bilirubin level $\leq$ 1.5 mg/dL, and aspartate transaminase [AST] level and alanine transaminase [ALT] levels $\leq$ ULN×2.5), renal

function (creatinine level $\leq 1.2$ mg/dL and creatinine clearance $\geq 60$ mL/min), and (5) absence of severe complications. In order to focus on evaluating the treatment efficacy of S-1, patients who underwent elective neck dissection and sentinel lymph node biopsies were excluded [19]. In addition, patients who previously received systemic therapy or radiotherapy and had distant metastasis, concomitant malignancies, active inflammatory disease, active gastric/duodenal ulcers, severe heart disease, or other severe concurrent disease were excluded. Pregnant or lactating women were also excluded.

## Treatment

After the completion of curative surgery, patients who consented to undergo S-1 adjuvant chemotherapy were assigned to the S-1 group, and those who refused S-1 adjuvant chemotherapy were assigned to the SA group. In the S-1 group, patients received 80 mg/day (body surface area [BSA] $< 1.25$ m$^2$), 100 mg/day (BSA $\geq 1.25$ to $< 1.5$ m$^2$), or 120 mg/day (BSA $\geq 1.5$ m2) of S-1, in two divided doses, daily, for 2 weeks, followed by a 1-week period of rest [11]. Administration of S-1 was started within 8 weeks after surgery, and the duration of treatment was 1 year. If adverse events meeting the criteria for temporary treatment withdrawal occurred, treatment was discontinued and was resumed when the criteria for treatment resumption were satisfied. If adverse events meeting the criteria for dose reduction developed, the dose was reduced by one level before treatment was resumed.

## Follow-up evaluations

After being allocated to the appropriate treatment group, the patients were followed-up for the evaluation of tumor control. We recorded local recurrence of the tumor, regional lymph node metastasis, and distant metastasis as local, regional, and distant failure, respectively. In patients with failed tumor control, we considered salvage surgery, radiotherapy, and/or additional chemotherapy. The survival after treatment was measured from the date of surgery to the date of death or last follow-up. The hematologic and non-hematologic toxicities of S-1 were prospectively scored according to the National Cancer Institute Common Toxicity Criteria for Adverse Events, version 3.0.

## Statistical analyses

The characteristics of the patients in the S-1 and SA groups were compared using Mann-Whitney's *U* and Fisher's exact tests for categorical and continuous factors, respectively, for the overall cohort (OC). For the propensity score-matched cohort (PSMC), the Wilcoxon signed-rank test was used for continuous factors and the exact McNemar test or stratified conditional logistic regression analysis was used for categorical factors.

The primary endpoint was disease-free survival (DFS), defined as the time from the date of surgery to the date of confirmation of recurrence, delayed cervical lymph node metastasis, distant metastasis, or the diagnosis of secondary cancer or death from any cause, whichever occurred first. The secondary endpoints were OS and safety. OS was defined as the time from the date of surgery to the date of death from any cause. The time to relapse (TTR) was defined as the time from surgery to the diagnosis of local recurrence or cervical lymph node metastasis. The OS and DFS were calculated using the Kaplan-Meier method, and the difference between the two groups was analyzed using the log-rank test. The hazard ratios (HRs) and 95% confidence intervals (CIs) were estimated by multivariate analyses, performed using the Cox proportional hazards regression model.

Prognostic and disease progression factors were considered for inclusion in the final models after calculating the coefficients and examining and ensuring that the proportion of missing

data was below 25% [20]. All factors showing significance on univariate analysis were considered to fit the model. All factors with $P < 0.2$ were reviewed to avoid missing important factors and were then examined using multivariate analysis [21]. Two-sided probabilities were used, and $P < 0.05$ was considered statistically significant, unless otherwise noted. The propensity score was calculated using a binary logistic regression that included the patients' characteristics. A propensity score, which reflected the probability of receiving S-1, was assigned to each patient. The S-1 and SA patients were randomly matched one-to-one, using greedy matching within propensity score calipers with no replacement [22]. The propensity scores were matched using a caliper width of 0.2 logits of the standard deviation to achieve a good covariate balance [22, 23]. The standardized differences were used to measure covariate balance, with an absolute standardized difference within 10% representing sufficient balance. The two matched subgroups were then analyzed for OS and DFS. Statistical analyses were performed using the Stata Statistical Software Program, Release 14.1 (StataCorp LP, College Station, TX, USA) and NCSS 10 Statistical Software Program (2015) (NCSS; LLC, Kaysville, UT, USA).

## Results

### Patient characteristics

From April 2004 to March 2012, a total of 95 cT2N0M0 OSCC patients were enrolled; 24 patients were assigned to the S-1 group and 71 to the SA group. The patient characteristics are summarized in Table 1. The only characteristic that significantly differed between the two groups was age ($P < 0.001$, Table 1).

### Study treatments

The numbers of patients in the S-1 group who received the study treatment after 3, 6, and 12 months were 19 (79.2%), 17 (70.8%), and 14 (58.3%), respectively (Table 2). The reasons for discontinuing treatment in this group were the development of recurrence or metastasis in 2 (8.3%) patients and the physician's judgement (mainly because of the occurrence of adverse events) in 8 (33.4%) patients.

### Adverse events

Table 3 shows the all-grade adverse events that occurred at an incidence rate of 4.2% (1 patient) or higher. An increase in the total bilirubin level was observed in 11 (45.8%) patients, anorexia was noted in 10 (41.7%) patients, anemia in 9 (37.5%) patients, fatigue and weight loss in 8 (33.3%) patients, thrombocytopenia in 7 (29.2%) patients, leukopenia, AST level increase and hyperpigmentation in 6 (25.0%) patients, rash/desquamation in 5 (20.8%) patients, ALT level increase in 4 (16.7%) patients, nausea in 3 (12.5%) patients, and vomiting in 2 (8.3%) patients. The following adverse events occurred at a severity of grade 3: anorexia in 2 (8.3%) patients and an increase in the total bilirubin level in 1 (4.2%) patient; they were in the S-1 group (Table 3). There were no treatment-related deaths in the S-1 group.

### Survival analyses in the OC

In the OC, 71 patients received SA and 24 received S-1. Although there were no significant differences, the S-1 group showed a better TTR, DFS, and OS than the SA group (Figs 1, 2A and 2B). In particular, the DFS was better in the S-1 group (Fig 2A).

**Table 1. Patient characteristics in the overall cohort.**

| Characteristics | | Surgery alone n (%) | S-1 adjuvant n (%) | *P*-value | |
|---|---|---|---|---|---|
| | Total | 71 (15.3) | 24 (84.7) | | |
| **Age (years)** | | | | | |
| Median | 69 | 73 | 62 | < 0.001 | (a) |
| ≤ 65 | 33 | 18 (54.5) | 15 (45.5) | 0.002 | (b) |
| > 65 | 62 | 53 (85.5) | 9 (14.5) | | |
| **Sex** | | | | | |
| Male | 53 | 39 (73.6) | 14 (26.4) | 0.816 | (b) |
| Female | 42 | 32 (76.2) | 10 (23.8) | | |
| **Oral subsite** | | | | | |
| Tongue | 52 | 36 (69.2) | 16 (30.8) | 0.347 | (b) |
| Maxilla | 12 | 11 (91.3) | 1 (8.3) | | |
| Mandible | 22 | 18(81.8) | 4 (18.2) | | |
| Oral floor | 2 | 1 (50.0) | 1 (50.0) | | |
| Buccal mucosa | 7 | 5 (71.4) | 2 (28.6) | | |
| **Clinical phenotype** | | | | | |
| Superficial | 41 | 32 (78.0) | 9 (22.0) | 0.519 | (b) |
| Exophytic | 23 | 15 (65.2) | 8 (34.8) | | |
| Endophytic | 31 | 24 (77.2) | 7 (22.6) | | |
| **Differentiation** | | | | | |
| Grade I | 68 | 51 (75.0) | 17 (25.0) | 0.885 | (b) |
| Grade II | 25 | 18 (72.0) | 7 (28.0) | | |
| Grade III | 2 | 2 (100.0) | 0 (0) | | |
| **Mode of invasion** | | | | | |
| I, II | 31 | 24 (77.4) | 7 (22.6) | 0.237 | (b) |
| III | 47 | 37 (78.7) | 10 (21.3) | | |
| IVc, IVd | 17 | 10 (58.8) | 7 (41.2) | | |
| **Local recurrence** | | | | | |
| No | 83 | 62 (74.7) | 21 (25.3) | > 0.999 | (b) |
| Yes | 12 | 9 (75.0) | 3 (25.0) | | |
| **Delayed cervical lymph node metastasis** | | | | | |
| No | 79 | 57 (72.2) | 22 (27.8) | 0.343 | (b) |
| Yes | 16 | 14 (87.5) | 2 (12.5) | | |
| **Distant metastasis** | | | | | |
| No | 92 | 69 (75.0) | 23 (25.0) | 0.384 | (b) |
| Yes | 3 | 2 (66.7) | 1 (33.3) | | |

(a) Mann-Whitney's *U* test for continuous factors and

(b) Fisher's exact test for categorical factors were used to calculate *P*-values between treatment options and clinicopathologic factors in 95 OSCC patients.

** indicated *P* < 0.01.

Abbreviations, OSCC, oral squamous cell carcinoma

## Survival analyses in the PSMC

In the PSMC, 20 patients each from the S-1 and SA groups were subjected to analysis after one-to-one propensity score matching (Table 4). As shown in Table 4, all baseline characteristics of patients in the PSMC were well-balanced ($P \geq 0.05$). The TTR of the S-1 group was significantly lower than that of the SA group ($P = 0.047$). In the S-1 group, a significant improvement in prognoses was observed with respect to the DFS ($P = 0.047$), but not with

**Table 2. Treatment completion rates with S-1 adjuvant chemotherapy.**

| | S-1 (n = 24) | |
|---|---|---|
| Duration | N | (%) |
| 3 months | 19 | (79.2) |
| 6 months | 17 | (70.8) |
| 12 months | 14 | (58.3) |

respect to the OS (*P* = 0.073; Figs 3, 4A and 4B). Although there was no statistical significance, the proportion of patients with cervical lymph node metastasis in the S-1 group tended to be smaller than that in the SA group (Table 4).

## Explanatory data analysis

In the OC, local recurrence developed in 9 patients in the SA group and 3 patients in the S-1 group. The HR was 1.249 (95% CI; 0.3377–4.616; Fig 5A). Delayed cervical lymph node metastasis developed in 14 patients in the SA group and 2 patients in the S-1 group. The HR was 2.809 (95% CI; 0.6377–12.38; Fig 5B). In the OC, comparison of the survival time from local recurrence or delayed cervical lymph node metastasis between the treatment groups revealed that the HR for death was 4.271 (95% CI; 0.5411–32.680) in the SA group compared to the S-1 group (Fig 5C). In contrast, in the PSMC, local recurrence developed in 4 patients in the SA group and 3 patients in the S-1 group. The HR was 1.869 (95% CI; 0.4157–8.404; Fig 6A). Delayed cervical lymph node metastasis developed in 5 patients in the SA group and 2 patients in the S-1 group. The HR was 3.191 (95% CI; 0.6152–16.55; Fig 6B). In the PSMC, comparison of the survival time from local recurrence or delayed cervical lymph node metastasis between

**Table 3. Adverse events with S-1 adjuvant chemotherapy.**

| | S-1 (n = 24) | | | |
|---|---|---|---|---|
| Adverse events | All grade | | Garde 3+4 | |
| | n | (%) | n | (%) |
| Leukopenia | 6 | (25.0) | 0 | (0.0) |
| Neutropenia | 1 | (4.2) | 0 | (0.0) |
| Thrombocytopenia | 7 | (29.2) | 0 | (0.0) |
| Anemia | 9 | (37.5) | 0 | (0.0) |
| Total bilirubin increase | 11 | (45.8) | 1 | (4.2) |
| AST increase | 6 | (25.0) | 0 | (0.0) |
| ALT increase | 4 | (16.7) | 0 | (0.0) |
| Fatigue | 8 | (33.3) | 0 | (0.0) |
| Anorexia | 10 | (41.7) | 2 | (8.3) |
| Weight loss | 8 | (33.3) | 0 | (0.0) |
| Rash/desquamation | 5 | (20.8) | 0 | (0.0) |
| Hyperpigmentation | 6 | (25.0) | 0 | (0.0) |
| Diarrhea | 1 | (4.2) | 0 | (0.0) |
| Mucositis/Stomatitis | 0 | (0.0) | 0 | (0.0) |
| Nausea | 3 | (12.5) | 0 | (0.0) |
| Vomiting | 2 | (8.3) | 0 | (0.0) |

Abbreviations: AST, aspartate transaminase and ALT, alanine transaminase

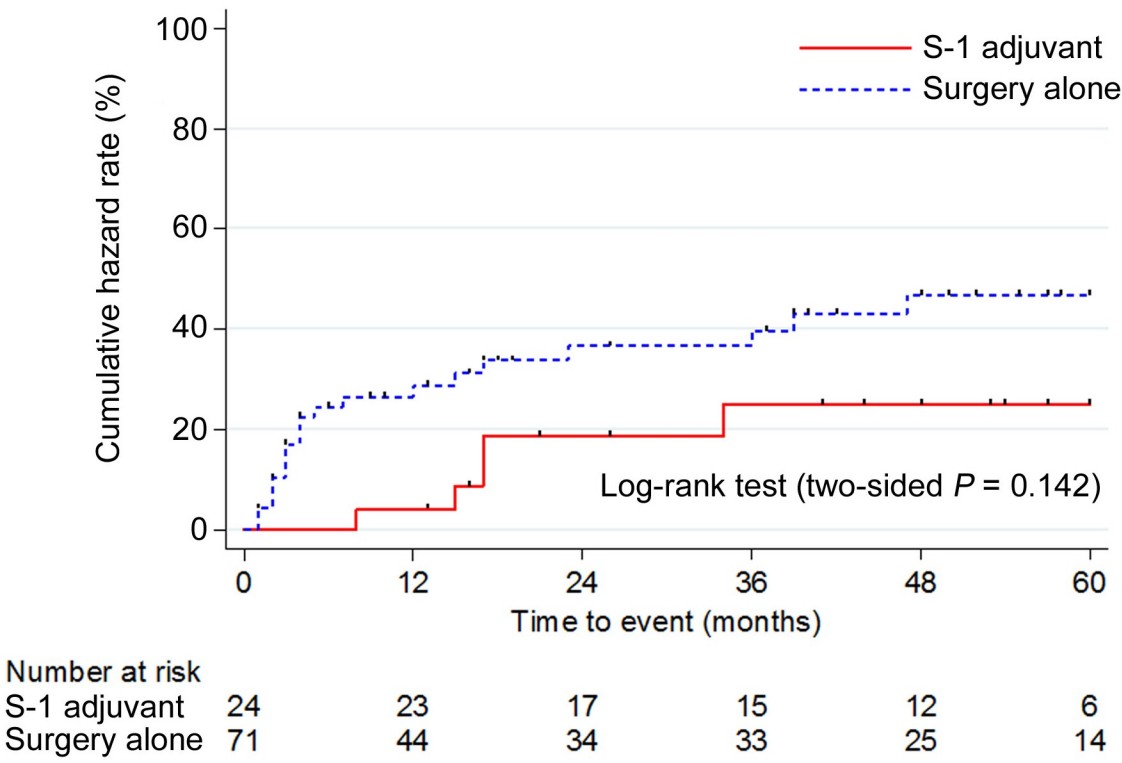

**Fig 1. Cumulative hazard rate of the time to relapse in the overall cohort.** TTR, time to relapse; OC, overall cohort.

the treatment groups indicated that the HR for death was 5.691 (95% CI; 0.6637–48.80) in the SA group compared to the S-1 group (Fig 6C).

## Discussion

This study was designed to evaluate the efficacy of adjuvant chemotherapy after curative surgery in patients with stage II OSCC. Tsukahara et al. recently reported convincing evidence for the benefits of adjuvant chemotherapy in patients with advanced head and neck cancer, who underwent curative therapy including surgery, radiotherapy, and chemoradiotherapy [13]. Therefore, we believe that investigating the effects of S-1 adjuvant chemotherapy in patients with early-stage OSCC may be valuable for establishing new treatment strategies. To the best of our knowledge, the present study is the first to indicate that adjuvant chemotherapy with S-1 improved the DFS in patients with stage II OSCC who received curative surgery for only the primary tumor compared to the DFS in a control group. Recently, Luryi et al. reported that in population-level data analyses, adjuvant chemotherapy is associated with compromised survival in patients with early-stage OSCC [24]. The study did not provide a detailed description of the chemotherapy regimens and included patients who underwent elective neck dissection; therefore, a detailed analysis of the differences in the results between this study and our study was not possible. However, it is necessary to understand the results of these studies and to interpret them carefully. The differences in the results may be related to the characteristic pharmacological action of S-1, as described below.

The treatment completion rate in the S-1 group was 58.3%. This rate was higher than the rate of 43.4% observed in a phase III study (ACTS-HNC) among patients with advanced

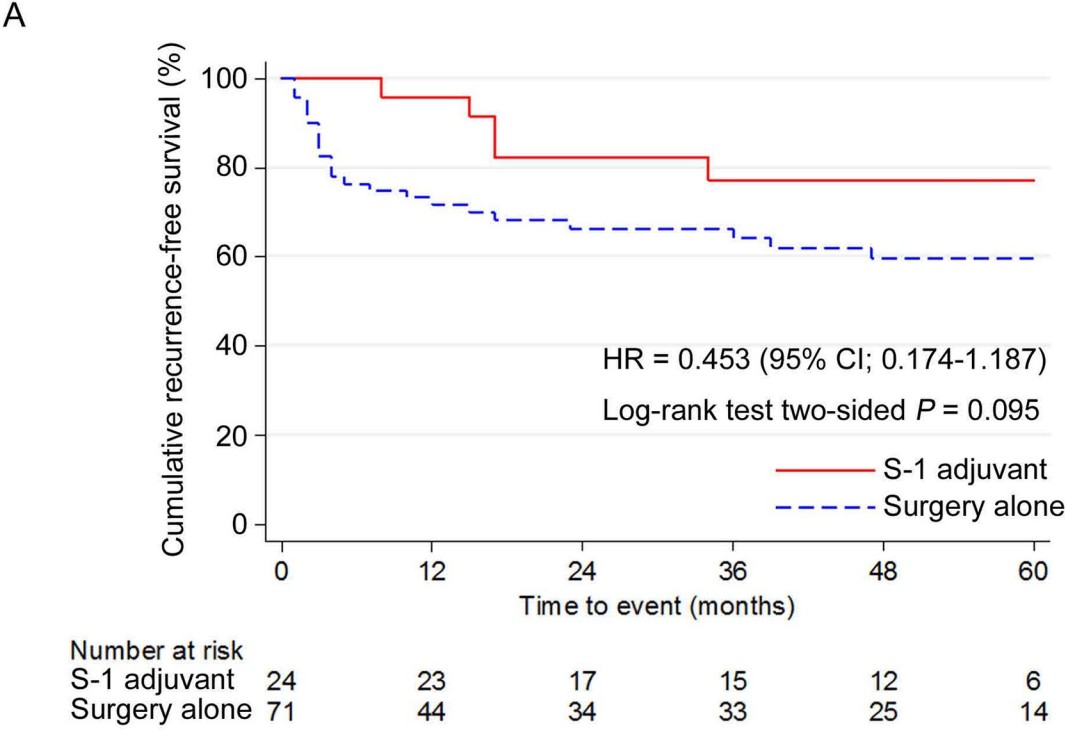

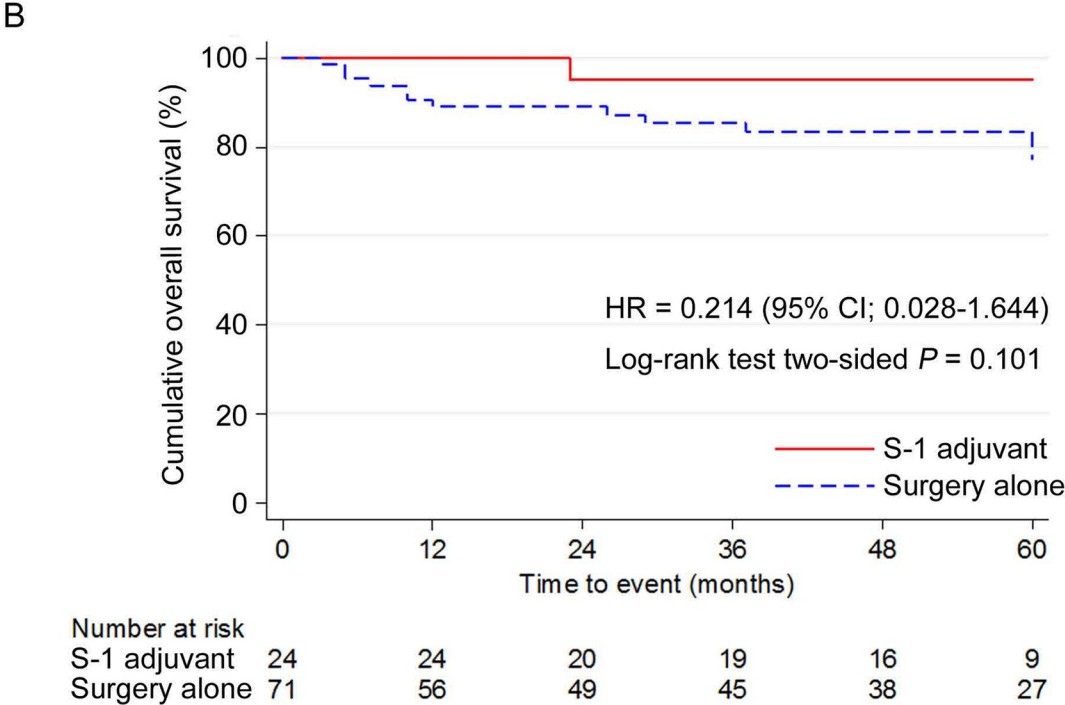

**Fig 2. Cumulative survival curves of the S-1 adjuvant therapy (S-1 adjuvant) and surgery alone (surgery alone) groups in the overall cohort.** (A) Disease-free survival. (B) Overall survival. DFS, Disease-free survival; OS, overall survival; OC, overall cohort.

**Table 4. Patient characteristics in the propensity score-matched cohort.**

| Characteristics | | Surgery alone n (%) | S-1 adjuvant n (%) | P—value | |
|---|---|---|---|---|---|
| | Total | 20 (50.0) | 20 (50.0) | | |
| **Age (years)** | | | | | |
| Median | 64.0 | 64.0 | 64.0 | 0.674 | (a) |
| ≤ 65 | 22 | 11 (50.0) | 11 (50.0) | > 0.999 | (b) |
| > 65 | 18 | 9 (50.0) | 9 (50.0) | | |
| **Sex** | | | | | |
| Male | 25 | 13 (52.0) | 12 (48.0) | > 0.999 | (b) |
| Female | 15 | 7 (46.7) | 8 (53.3) | | |
| **Oral subsite** | | | | | |
| Tongue | 23 | 10 (43.5) | 13 (56.5) | 0.573 | (c) |
| Maxilla | 3 | 2 (66.7) | 1 (33.3) | | |
| Mandible | 10 | 6 (60.0) | 4 (40.0) | | |
| Oral floor | 1 | 1 (100.0) | 0 (0.0) | | |
| Buccal mucosa | 3 | 1 (33.3) | 2 (66.7) | | |
| **Clinical phenotype** | | | | | |
| Superficial | 16 | 8 (50.0) | 8 (50.0) | 0.698 | (c) |
| Exophytic | 10 | 4 (40.0) | 6 (60.0) | | |
| Endophytic | 14 | 8 (57.1) | 6 (42.9) | | |
| **Differentiation** | | | | | |
| Grade I | 30 | 16 (53.3) | 14 (46.7) | 0.754 | (b) |
| Grade II | 10 | 4 (40.0) | 6 (60.0) | | |
| **Mode of invasion** | | | | | |
| I, II | 13 | 6 (46.2) | 7 (53.8) | 0.517 | (c) |
| III | 19 | 12 (63.2) | 7 (36.8) | | |
| IVc, IVd | 8 | 2 (25.0) | 6 (74.0) | | |
| **Local recurrence** | | | | | |
| No | 33 | 16 (48.5) | 17 (51.5) | > 0.999 | (b) |
| Yes | 7 | 4 (57.1) | 3 (42.9) | | |
| **Delayed cervical lymph node metastasis** | | | | | |
| No | 33 | 15 (45.5) | 18 (54.5) | 0.453 | (b) |
| Yes | 7 | 5 (71.4) | 2 (28.6) | | |
| **Distant metastasis** | | | | | |
| No | 38 | 19 (50,0) | 19 (50.0) | 1.000 | (b) |
| Yes | 2 | 1 (50.0) | 1 (50.0) | | |

(a) Wilcoxon signed-rank test for continuous factors,

(b) Exact McNemar test for 2 x 2 categorical factors and

(c) Stratified conditional logistic regression for 2 x m categorical factors were used to calculate P-values between treatment options and clinicopathologic factors in 40 OSCC patients.

HNSCC [13]. This difference may be attributed to the decrease in residual function in patients with advanced HNSCC, who underwent definitive therapy. In view of these findings, clinicians should carefully consider both, hematologic and non-hematological toxicities, and provide supportive therapy to prevent the discontinuation of S-1. However, the low incidence rates of grade 3 or higher grade adverse events in our study support the notion that S-1 administration may be an acceptable treatment option to further improve the prognoses of patients with stage II OSCC.

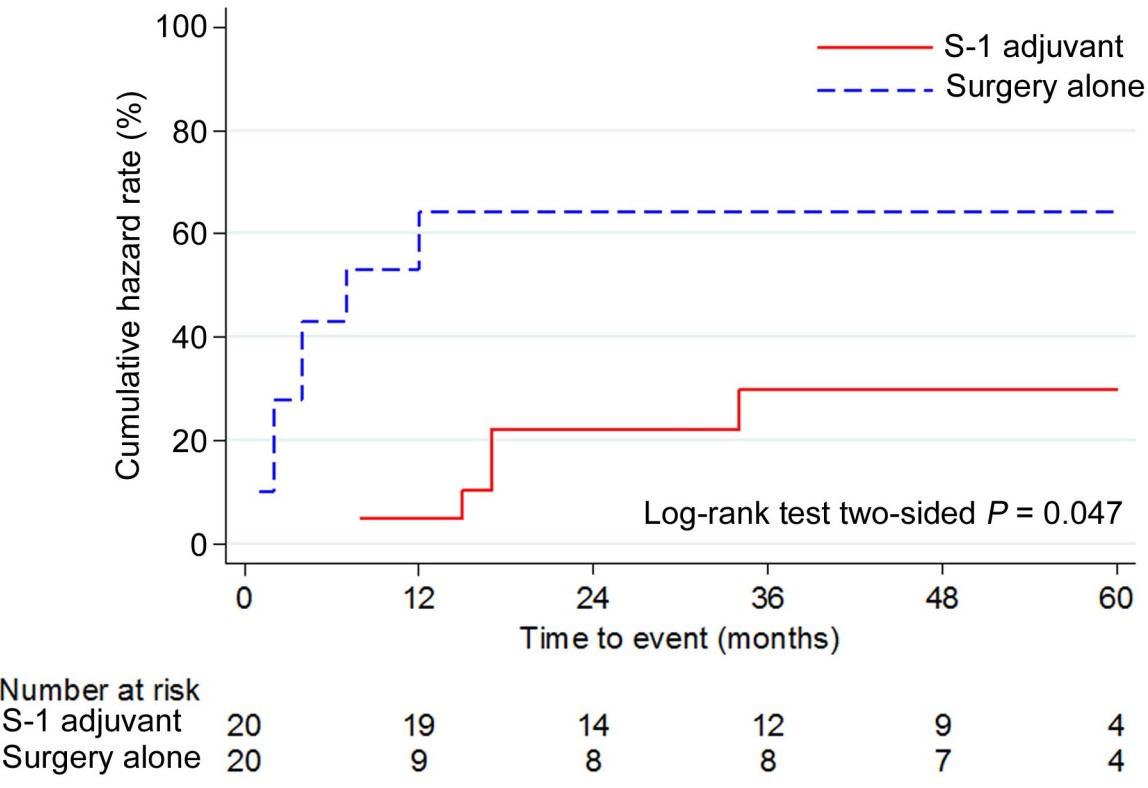

| Number at risk | | | | | | |
|---|---|---|---|---|---|---|
| S-1 adjuvant | 20 | 19 | 14 | 12 | 9 | 4 |
| Surgery alone | 20 | 9 | 8 | 8 | 7 | 4 |

**Fig 3. Cumulative hazard rate of the time to relapse in the propensity score-matched cohort.** PSMC, propensity score-matched cohort; TTR, time to relapse.

It is not clear why patient survival in the S-1 group was better than that in the SA group. Although there was no statistically significant difference between the groups, the present data, including the results of explanatory data analyses, showed that the cumulative rates of local recurrence and delayed cervical lymph node metastasis in the S-1 group tended to be smaller than those in the SA group in both the OC and PSMC. In addition, the time from recurrence or delayed cervical lymph node metastasis to death tended to be longer in the S-1 group. As observed in a previous study (ACTS-HNC) [13], these results possibly indicate that S-1 contributes to disease control after loco-regional failure in patients with OSCC. Among the various clinicopathological characteristics, local recurrence and/or regional lymph node metastasis have been proposed to be the prognostic indicators following surgery in patients with OSCC [25, 26]. Tumor angiogenesis is a hallmark of cancer; it is the essential process underlying tumor growth and progression, and, thereby, contributes to recurrence or metastasis. However, S-1 and its metabolites have been shown to suppress angiogenesis [27–29]. The anti-angiogenic effect of chemotherapy is known to be optimized through the metronomic administration of such drugs for prolonged periods [30]. Collectively, the survival benefit of S-1 administration in this study was probably attributable to both, the cytotoxic and anti-angiogenic activities.

Among early-stage OSCC patients, END has been shown to result in higher survival rates than therapeutic neck dissection [31]. However, END results in overtreatment in more than 70% of early OSCC patients and a high rate of complications [32]. In order to resolve these problems, sentinel lymph node biopsy (SLNB), which is less invasive and improves patients' quality of life, has gained popularity in the treatment of patients with early-stage OSCC [33–

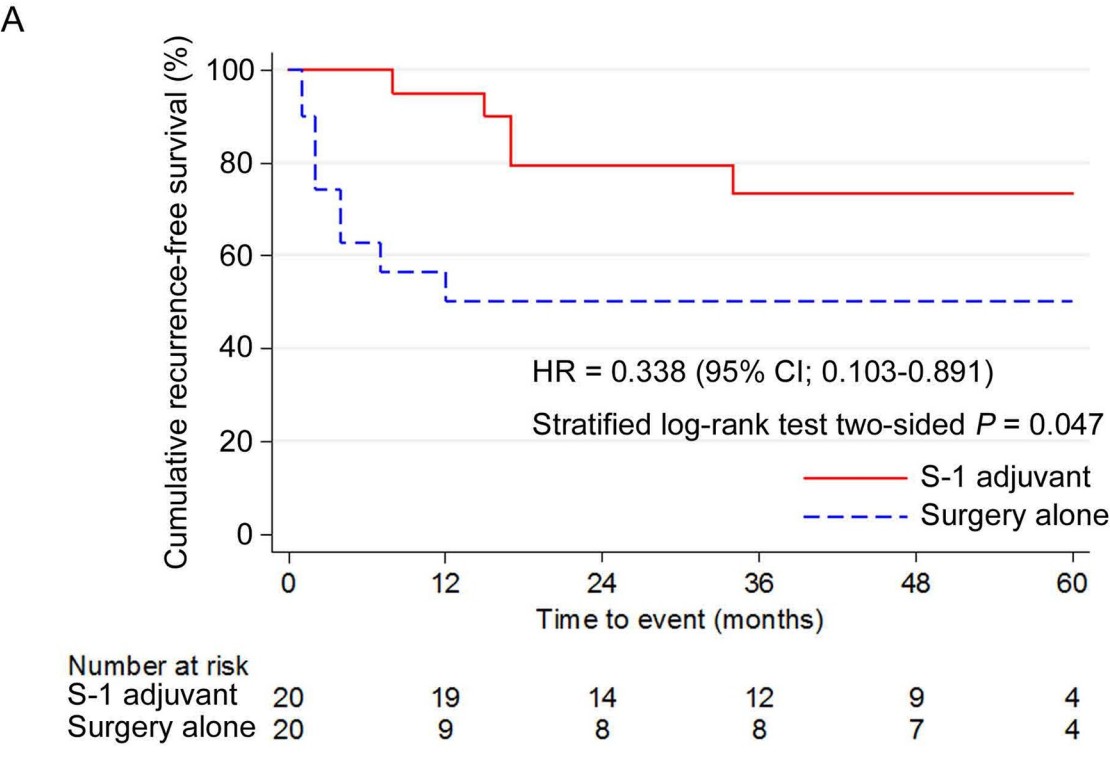

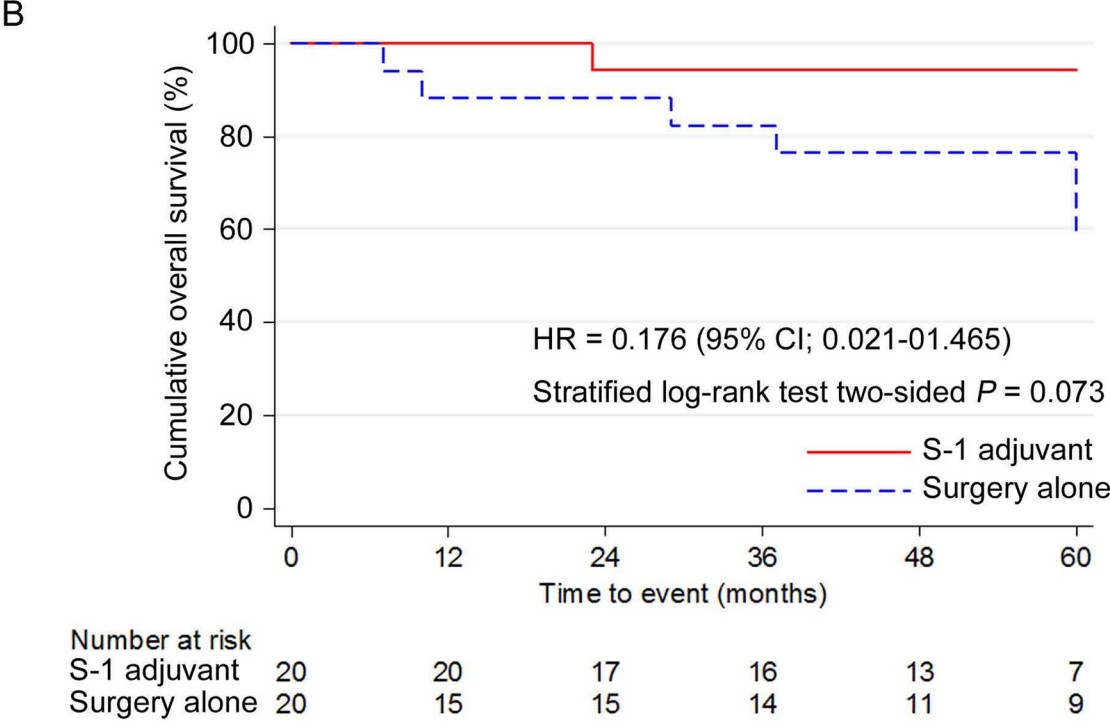

**Fig 4. Cumulative survival curves of the S-1 adjuvant therapy (S-1 adjuvant) and surgery alone (surgery alone) groups in the propensity score-matched cohort.** (A) Disease-free survival. (B) Overall survival. DFS, Disease-free survival; OS, overall survival PSMC, propensity score-matched cohort.

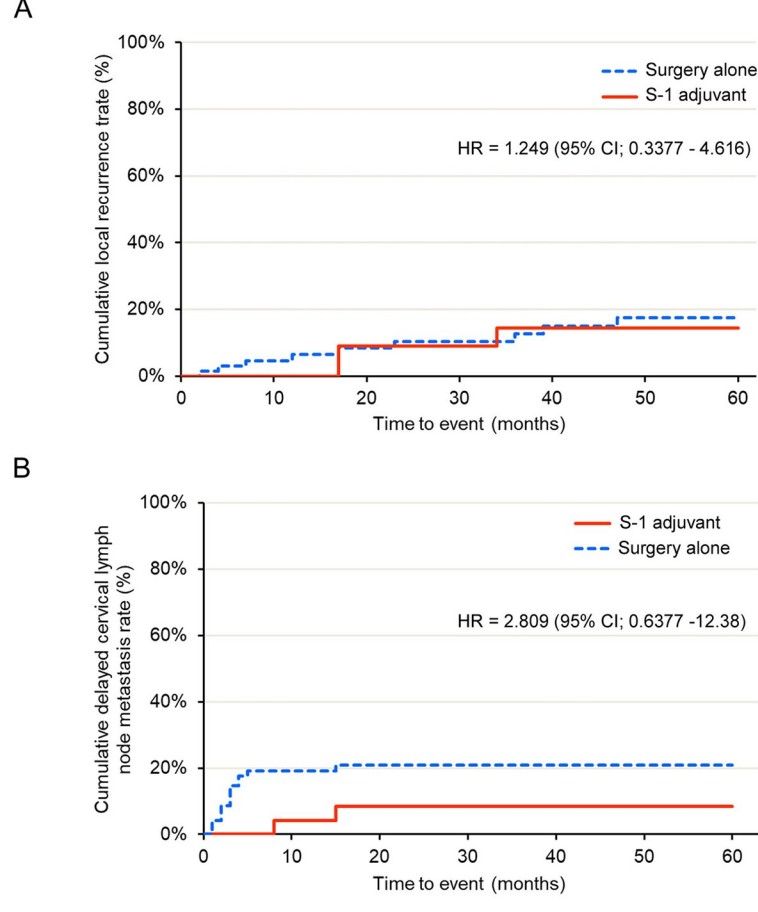

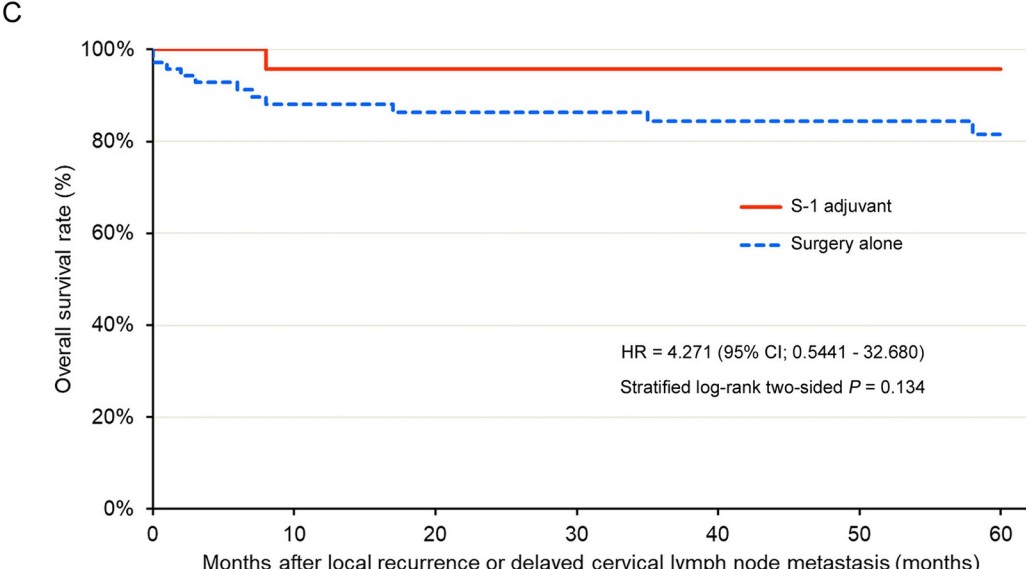

**Fig 5. Explanatory data analysis in overall cohort.** (A) Cumulative local recurrence rate of the S-1 adjuvant therapy (S-1 adjuvant) and surgery alone (surgery alone) groups. (B) Cumulative delayed cervical lymph node metastasis rate of the S-1 adjuvant therapy (S-1 adjuvant) and surgery alone (Surgery alone) groups. (C) Survival from loco-regional failures to death in patients with local recurrence/ delayed cervical lymph node metastasis.

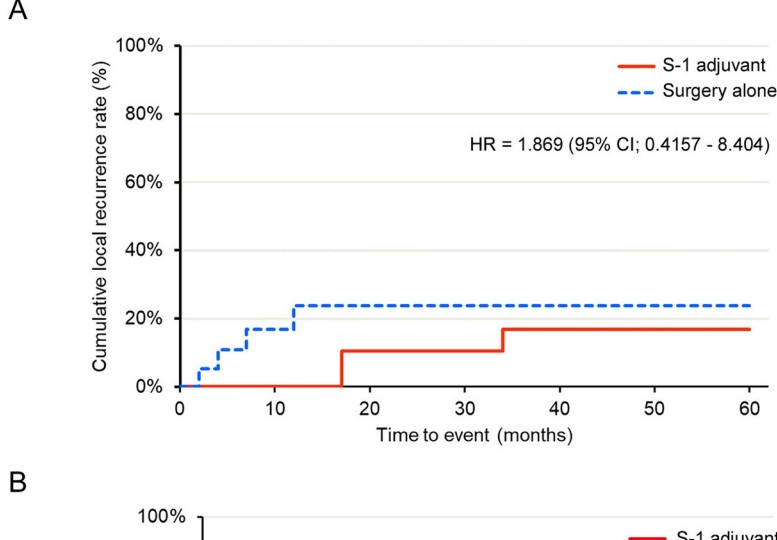

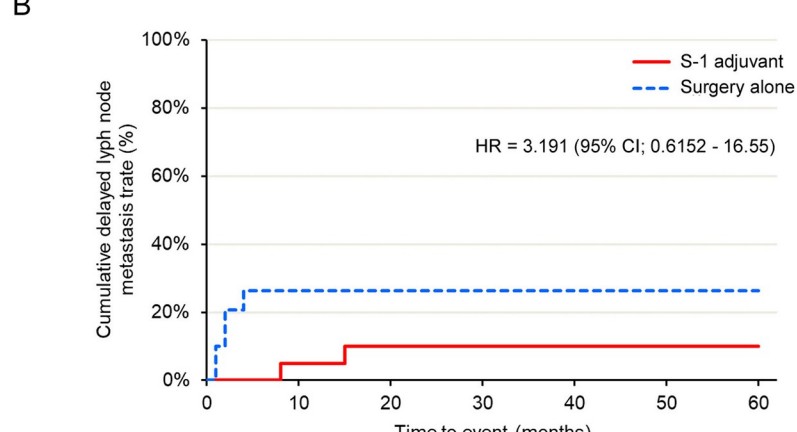

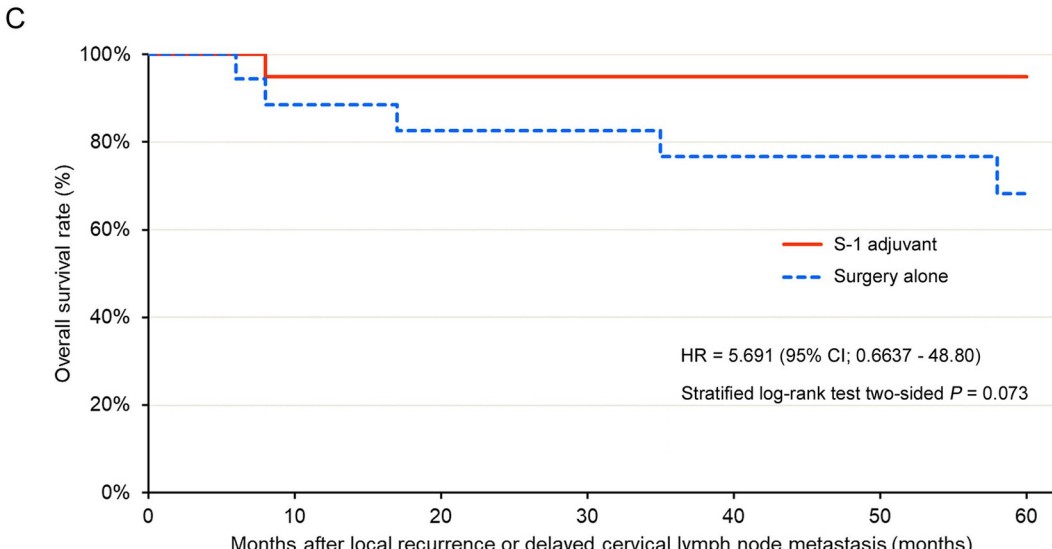

**Fig 6. Explanatory data analysis in the propensity score-matched cohort.** (A) Cumulative local recurrence rate of the S-1 adjuvant therapy (S-1 adjuvant) and surgery alone (surgery alone) groups. (B) Cumulative delayed cervical lymph node metastasis rate of the S-1 adjuvant therapy (S-1 adjuvant) and surgery alone (surgery alone) groups. (C) Survival from loco-regional failures to death in patients with local recurrence/ delayed cervical lymph node metastasis.

36]. However, SLNB may not be universally applicable in routine medical practice. Therefore, in addition to SLNB, S-1 may warrant consideration as a therapeutic option in the cervical management of patients with early OSCC who undergo curative resection only for the primary tumor.

A limitation associated with our study is the small sample size; further studies with larger sample sizes are required to confirm the superiority of S-1 adjuvant chemotherapy over SA. In addition, comparative studies with other treatment options should be considered to confirm the superiority of S-1.

In conclusion, this retrospective study suggests that S-1 therapy was more effective than SA in the PSMC. We believe that S-1 adjuvant chemotherapy followed by curative surgery should be considered the standard of care in future phase III trials including patients with stage II (T2N0) OSCC.

## Acknowledgments

We would like to thank Editage for English language editing.

## Author Contributions

**Conceptualization:** Ryoji Yoshida, Akimitsu Hiraki, Masanori Shinohara, Hideki Nakayama.

**Data curation:** Ryoji Yoshida, Akiyuki Hirosue, Masatoshi Hirayama, Yuichiro Matsuoka, Junki Sakata, Ken Kikuchi.

**Formal analysis:** Ryoji Yoshida, Masashi Nagata, Ken Kikuchi.

**Investigation:** Kenta Kawahara, Masafumi Nakamoto, Nozomu Takahashi, Hikaru Nakashima, Hidetaka Arita.

**Methodology:** Ken Kikuchi.

**Project administration:** Akimitsu Hiraki, Masanori Shinohara, Hideki Nakayama.

**Writing – original draft:** Ryoji Yoshida, Masashi Nagata, Ken Kikuchi.

**Writing – review & editing:** Akimitsu Hiraki, Masanori Shinohara, Hideki Nakayama.

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
