## [Decision Letter · Decision Letter 0]

20 Jan 2020

PONE-D-19-27210

Efficacy of adjuvant chemotherapy with S-1 in stage II oral squamous cell carcinoma patients: A comparative study using the propensity score matching method

PLOS ONE

Dear D.D.S.,PhD. Yoshida,

Thank you for submitting your manuscript to PLOS ONE. After careful consideration, we feel that it has merit but does not fully meet PLOS ONE’s publication criteria as it currently stands. Therefore, we invite you to submit a revised version of the manuscript that addresses the points raised during the review process.

ACADEMIC EDITOR:  Although the case number is too small, the RFS benefits are still noted. I suggest the authors validate the findings in an independent cohort soon. Of course, a phase III study would possibly change the current treatment guideline.  In addition, a 30% recurrence rate was observed in the first year after surgery (Figure 2.A), which seems too high. Please confirm that. 

We would appreciate receiving your revised manuscript by Mar 05 2020 11:59PM. To enhance the reproducibility of your results, we recommend that if applicable you deposit your laboratory protocols in protocols.io, where a protocol can be assigned its own identifier (DOI) such that it can be cited independently in the future. For instructions see: http://journals.plos.org/plosone/s/submission-guidelines#loc-laboratory-protocols

We look forward to receiving your revised manuscript.

Kind regards,

Jason Chia-Hsun Hsieh, M.D. Ph.D

Academic Editor

PLOS ONE

Journal Requirements:

2. We noticed you have some minor occurrence(s) of overlapping text with the following previous publication(s), which needs to be addressed:

https://doi.org/10.1016/j.ijrobp.2009.03.055

https://doi.org/10.1002/cam4.476

https://doi.org/10.1371/journal.pone.0116965

In your revision ensure you cite all your sources (including your own works), and quote or rephrase any duplicated text outside the Methods section. Further consideration is dependent on these concerns being addressed.

Additional Editor Comments:

Although the case number is too small, the RFS benefits are still noted. I suggest the authors validate the findings in an independent cohort soon. Of course, a phase III study would possibly change the current treatment guideline. In addition, a 30% recurrence rate was observed in the first year after surgery (Figure 2.A), which seems too high. Please confirm that.

Reviewers' comments:

Reviewer's Responses to Questions

**Comments to the Author**

1. Is the manuscript technically sound, and do the data support the conclusions?

Reviewer #1: Partly

Reviewer #2: Partly

Reviewer #3: Partly

2. Has the statistical analysis been performed appropriately and rigorously? 

Reviewer #1: Yes

Reviewer #2: Yes

Reviewer #3: Yes

3. Have the authors made all data underlying the findings in their manuscript fully available?

Reviewer #1: No

Reviewer #2: Yes

Reviewer #3: Yes

4. Is the manuscript presented in an intelligible fashion and written in standard English?

Reviewer #1: Yes

Reviewer #2: No

Reviewer #3: Yes

5. Review Comments to the Author

Reviewer #1: In the manuscript entitled ‘Efficacy of adjuvant chemotherapy with S-1 in stage II oral squamous cell carcinoma patients: A comparative study using the propensity score matching method’, Yoshida et al. evaluated the efficacy and safety of S-1 adjuvant chemotherapy compared with surgery alone in patients with stage II oral squamous cell carcinoma.

The effects of S1 adjuvent chemotherapy have been studied in multiple head and neck cancer types and in this regard, the current manuscript is incremental over our current knowledge of the effects of S1 adjuvent chemotherapy in SCC. To make the study suitable for publication in PLOS ONE, some qualitative and quantitative analysis of the lesions from follow up evaluations would be necessary. The authors mentioned that tumor local recurrence, regional lymph node metastasis, and distant metastasis as local, regional and distant failure were recorded. Some visual evidence in support of the same, along with graphical analysis would make the study more resourceful and suitable for publication in this journal.

In its current form, the study is more suitable for specialist clinical journals.

Reviewer #2: The authors have undertake a comparative study using the propensity score matching method for the efficacy of adjuvant chemotherapy with S-1 in stage II oral squamous cell carcinoma patients. The efficacy of adjuvant S-1 therapy In oral squamous cell carcinoma has not been reported yet, so this paper is very significant.

#1

In eligibility criteria, why creatine level ≥ 1.2 ?

#2

Is there any difference of surgical way between S1 and SA ? If so, you should consider the difference.

Reviewer #3: 1.The eligibility criteria in the study were patients with cT2N0 oral cancer, but cervical lymph node metastases were found in both the S-1 and the two groups of patients undergoing surgery alone (Table 1). It did not seem to meet the original eligibility criteria. Use of RT correlates with statistically significantly improved overall survival and cause-specific survival in patients with T2 disease. And the results did not specify the rate of neck lymph node dissection among patients undergoing surgery. In previous study, adjuvant RT significantly improves overall survival for patients with node-positive HNSCC. If these two groups of patients were node positive, adjuvant radiotherapy is required after surgery.

2. A large restrospective analysis have shown that adjuvant chemotherapy in patients with oral squamous cell carcinoma is associated with reduced survival(Luryi AL, et al. JAMA Otolaryngol Head Neck Surg. 2015. In current study showed that S-1 adjuvant chemotherapy maybe more efficient than surgery alone in early-stage OSCC patients. How to explain the differences between these finding.

3. In this retrospective study, the sample size of patients in the S-1 group was too small. Testing the efficacy of adjuvant chemotherapy with a limited number is statistically too ineffective.

6. PLOS authors have the option to publish the peer review history of their article (what does this mean?). If published, this will include your full peer review and any attached files.

Reviewer #1: No

Reviewer #2: No

Reviewer #3: No

---

## [Author Response · Author response to Decision Letter 0]

19 Feb 2020

Responses to the Reviewers’ comments

Responses to Academic Editor

We sincerely appreciate the helpful comments from the reviewers, which have provided further insight and have helped considerably in improving the paper. As summarized below, we have revised the manuscript extensively, based on the reviewers’ comments. In the revised version of our paper, all of the changes are underlined and highlighted in yellow.

Comments: Although the case number is too small, the RFS benefits are still noted. I suggest the authors validate the findings in an independent cohort soon. Of course, a phase III study would possibly change the current treatment guideline. In addition, a 30% recurrence rate was observed in the first year after surgery (Figure 2.A), which seems too high. Please confirm that.

Comment 1: Although the case number is too small, the RFS benefits are still noted. I suggest the authors validate the findings in an independent cohort soon. Of course, a phase III study would possibly change the current treatment guideline.

Response to Comment 1: 

We sincerely appreciate the pertinent observations and suggestions. However, owing to the limited duration of the study and the number of cases, we were unable to collect enough data for validation. We fully understand the need for validation; however, it was not feasible at present. We have added an exploratory analysis (Figures 5 and 6), based on the comments from other reviewers, and have revised the content of the Discussion section. We have added the following text in the Results section:

Page 13, line 220–Page 14, line 233

“Explanatory Data Analysis

In the OC, local recurrence developed in 9 patients in the SA group and 3 patients in the S-1 group. The HR was 1.249 (95% CI; 0.3377–4.616; Fig. 6A). Delayed cervical lymph node metastasis developed in 14 patients in the SA group and 2 patients in the S-1 group. The HR was 2.809 (95% CI; 0.6377–12.38; Fig. 6B). In the OC, comparison of the survival time from local recurrence or delayed cervical lymph node metastasis between the treatment groups revealed that the HR for death was 4.271 (95% CI; 0.5411–32.680) in the SA group compared to the S-1 group (Fig. 5C). In contrast, in the PSMC, local recurrence developed in 4 patients in the SA group and 3 patients in the S-1 group. The HR was 1.869 (95% CI; 0.4157– 8.404; Fig. 6A). Delayed cervical lymph node metastasis developed in 5 patients in the SA group and 2 patients in the S-1 group. The HR was 3.191 (95% CI; 0.6152–16.55; Fig. 6B). In the PSMC, comparison of the survival time from local recurrence or delayed cervical lymph node metastasis between the treatment groups indicated that the HR for death was 5.691 (95% CI; 0.6637–48.80) in the SA group compared to the S-1 group (Fig. 6C).”

We have made the following revisions in the Discussion section, as follows:

Page 15, lines 252- 261

“To the best of our knowledge, the present study is the first to indicate that adjuvant chemotherapy with S-1 improved the DFS in patients with stage II OSCC who received curative surgery for only the primary tumor compared to the DFS in a control group. Recently, Luryi et al. reported that in population-level data analyses, adjuvant chemotherapy is associated with compromised survival in patients with early-stage OSCC [24]. The study did not provide a detailed description of the chemotherapy regimens and included patients who underwent elective neck dissection; therefore, a detailed analysis of the differences in the results between this study and our study was not possible. However, it is necessary to understand the results of these studies and to interpret them carefully. The differences in the results may be related to the characteristic pharmacological action of S-1, as described below.”

Comment 2: In addition, a 30% recurrence rate was observed in the first year after surgery (Figure 2.A), which seems too high. Please confirm that.

Responses to Comment 2: 

We had initially used the phrase “relapse-free survival", which was misleading to the readers. In this study, patients who underwent elective neck dissection were excluded, and patients with “relapse” included those with delayed cervical lymph node metastasis. The description “relapse” provided the impression that the incidence of local recurrence was approximately 30%; the description has accordingly been revised from “relapse-free survival” to “disease-free survival”. We also revised the terms “loco-regional recurrence” to “local recurrence” and “cervical lymph node metastasis” to “delayed cervical lymph node metastasis”. In our department, approximately 20% of the cT2N0 patients have delayed cervical lymph node metastasis after primary tumor resection; in view of the local recurrence rate, the loco-regional failure of 30% may be considered a reasonable value. Finally, we have revised the Materials and Methods section as follows: 

Page 6, line 135–Page 7, line 137

“The primary endpoint was disease-free survival (DFS), defined as the time from the date of surgery to the date of confirmation of recurrence, delayed cervical lymph node metastasis, distant metastasis, or the diagnosis of secondary cancer or death from any cause, whichever occurred first.”

The following sentences were added in the Materials and Methods to clarify the eligibility criteria for this study:

Page 4, lines 79–82

“In our department, elective neck dissection is only performed for cases of N0 oral cancer for the purpose of reconstruction, and a “wait-and-see policy” has been adopted. Therefore, all patients enrolled in the present study only underwent resection of the primary tumor.”

Furthermore, the following text has been added in the Discussion section: 

Page 17, lines 291–293

"Therefore, in addition to SLNB, S-1 may warrant consideration as a therapeutic option in the cervical management of patients with early OSCC who undergo curative resection only for the primary tumor. “

Responses to Reviewer 1:

We sincerely appreciate your helpful comments, which have provided further insight and have helped considerably in improving the paper. As summarized below, we have revised the manuscript extensively. In the revised version of our paper, all of the changes are underlined and highlighted in yellow.

Comments: The effects of S1 adjuvent chemotherapy have been studied in multiple head and neck cancer types and in this regard, the current manuscript is incremental over our current knowledge of the effects of S1 adjuvent chemotherapy in SCC. To make the study suitable for publication in PLOS ONE, some qualitative and quantitative analysis of the lesions from follow up evaluations would be necessary. The authors mentioned that tumor local recurrence, regional lymph node metastasis, and distant metastasis as local, regional and distant failure were recorded. Some visual evidence in support of the same, along with graphical analysis would make the study more resourceful and suitable for publication in this journal.

Response to Comments: We appreciate your pertinent observations and helpful suggestions. We have added the incidence of distant metastasis in Tables 1 and 4, have performed an exploratory analysis, and have added Figures 5 and 6 in the revised manuscript.

The following information has been added in the Results section 

Page 13, line 220–Page 14, line 233

“Explanatory Data Analysis

In the OC, local recurrence developed in 9 patients in the SA group and 3 patients in the S-1 group. The HR was 1.249 (95% CI; 0.3377–4.616; Fig. 5A). Delayed cervical lymph node metastasis developed in 14 patients in the SA group and 2 patients in the S-1 group. The HR was 2.809 (95% CI; 0.6377–12.38; Fig. 5B). In the OC, comparison of the survival time from local recurrence or delayed cervical lymph node metastasis between the treatment groups revealed that the HR for death was 4.271 (95% CI; 0.5411–32.680) in the SA group compared to the S-1 group (Fig. 5C). In contrast, in the PSMC, local recurrence developed in 4 patients in the SA group and 3 patients in the S-1 group. The HR was 1.869 (95% CI; 0.4157– 8.404; Fig. 6A). Delayed cervical lymph node metastasis developed in 5 patients in the SA group and 2 patients in the S-1 group. The HR was 3.191 (95% CI; 0.6152–16.55; Fig. 6B). In the PSMC, comparison of the survival time from local recurrence or delayed cervical lymph node metastasis between the treatment groups indicated that the HR for death was 5.691 (95% CI; 0.6637–48.80) in the SA group compared to the S-1 group (Fig. 6C).”

The following text has been added in the Discussion section: 

Page 16, lines 271–277 

“Although there was no statistically significant difference between the groups, the present data, including the results of explanatory data analyses, showed that the cumulative rates of local recurrence and delayed cervical lymph node metastasis in the S-1 group tended to be smaller than those in the SA group in both the OC and PSMC. In addition, the time from recurrence or delayed cervical lymph node metastasis to death tended to be longer in the S-1 group. As observed in a previous study (ACTS-HNC) [13], these results possibly indicate that S-1 contributes to disease control after loco-regional failure in patients with OSCC.”

Responses to Reviewer 2:

Thank you for the valuable comments, which have provided further insight and have helped considerably in improving the paper. We have revised the manuscript extensively, as suggested. In the revised version of our paper, all of the changes are underlined and highlighted in yellow.

Comment #1: In eligibility criteria, why creatine level ≥ 1.2 ?

Response: As correctly observed, this part was incorrect. We have accordingly changed this to “creatinine level ≤ 1.2 mg/dL” in the revised manuscript (page 5, line 99).

Comment #2: Is there any difference of surgical way between S1 and SA ? If so, you should consider the difference.

Response: There was no difference in the surgical procedure between the SA and S-1 groups. We have added the following information in the Materials and Methods section for clarification:

Page 4, lines 79–82

“In our department, elective neck dissection is only performed for cases of N0 oral cancer for the purpose of reconstruction, and a “wait-and-see policy” has been adopted. Therefore, all patients enrolled in the present study only underwent resection of the primary tumor.”

Furthermore, the following text has been added in the Discussion section: 

Page 17, lines 291–293

"Therefore, in addition to SLNB, S-1 may warrant consideration as a therapeutic option in the cervical management of patients with early OSCC who undergo curative resection only for the primary tumor. “

Responses to Reviewer 3:

We thank you for the helpful comments, which have provided further insight and have helped considerably in improving the paper. As summarized below, we have revised the manuscript extensively in accordance with your comments. In the revised version of our paper, all of the changes are underlined and highlighted in yellow.

Comment #1: The eligibility criteria in the study were patients with cT2N0 oral cancer, but cervical lymph node metastases were found in both the S-1 and the two groups of patients undergoing surgery alone (Table 1). It did not seem to meet the original eligibility criteria. Use of RT correlates with statistically significantly improved overall survival and cause-specific survival in patients with T2 disease. And the results did not specify the rate of neck lymph node dissection among patients undergoing surgery. In previous study, adjuvant RT significantly improves overall survival for patients with node-positive HNSCC. If these two groups of patients were node positive, adjuvant radiotherapy is required after surgery.

Response: We had initially used the phrase “cervical lymph node metastasis”, which was misleading for the readers. In our department, elective neck dissection is only performed for cases of N0 oral cancer for the purpose of reconstruction, and a “wait-and-see policy” has been adopted. Therefore, patients who underwent elective neck dissection were excluded from this study. To avoid confusion, we revised the term “cervical lymph node metastasis” to “delayed cervical lymph node metastasis”. The following information has been added in the Materials and Methods section to clarify the eligibility criteria for this study:

Page 4, line 79–82

“In our department, elective neck dissection is only performed for cases of N0 oral cancer for the purpose of reconstruction, and a “wait-and-see policy” has been adopted. Therefore, all patients enrolled in the present study only underwent resection of the primary tumor.”

Moreover, the following text has been added in the Discussion section: 

Page 17, lines 291–293

"Therefore, in addition to SLNB, S-1 may warrant consideration as a therapeutic option in the cervical management of patients with early OSCC who undergo curative resection only for the primary tumor. “

Comment #2: A large restrospective analysis have shown that adjuvant chemotherapy in patients with oral squamous cell carcinoma is associated with reduced survival (Luryi AL, et al. JAMA Otolaryngol Head Neck Surg. 2015. In current study showed that S-1 adjuvant chemotherapy maybe more efficient than surgery alone in early-stage OSCC patients. How to explain the differences between these finding.

Response: As accurately observed, Luryi et al. recently reported that in population-level data analyses, adjuvant chemotherapy is associated with compromised survival in early-stage OSCC. The study does not provide a detailed description of the chemotherapy regimens and included patients who underwent elective neck dissection; therefore, it is not possible to analyze in detail why our results were contrary to those of Luryi et al. However, it is necessary to understand the results of such studies and to interpret them carefully. The difference may also be related to the characteristic pharmacological action of S-1. Therefore, we have added the following text in the Discussion section:

Page 15, lines 252-261

“To the best of our knowledge, the present study is the first to indicate that adjuvant chemotherapy with S-1 improved the DFS in patients with stage II OSCC who received curative surgery for only the primary tumor compared to the DFS in a control group. Recently, Luryi et al. reported that in population-level data analyses, adjuvant chemotherapy is associated with compromised survival in patients with early-stage OSCC [24]. The study did not provide a detailed description of the chemotherapy regimens and included patients who underwent elective neck dissection; therefore, a detailed analysis of the differences in the results between this study and our study was not possible. However, it is necessary to understand the results of these studies and to interpret them carefully. The differences in the results may be related to the characteristic pharmacological action of S-1, as described below.”

Comment #3: In this retrospective study, the sample size of patients in the S-1 group was too small. Testing the efficacy of adjuvant chemotherapy with a limited number is statistically too ineffective.

Response: We appreciate your concerns. However, owing to the limited duration of the study and the number of cases, we were unable to collect enough data. We fully understand that testing the efficacy of adjuvant chemotherapy with a limited number of cases is statistically ineffective. We had therefore, mentioned this as a limitation in the original manuscript as follows:

“A limitation associated with our study is the small sample size; further studies with larger sample sizes are required to confirm the superiority of S-1 adjuvant chemotherapy over SA.”

Nevertheless, we performed additional detailed analyses using current data. We have added the results of the exploratory analysis (Figures 5 and 6), in accordance with the comments from the other reviewers, and have revised the discussion accordingly. We have added the following information in the Results section:

Page 13, line 220–Page 14, line 233

“Explanatory Data Analysis

In the OC, local recurrence developed in 9 patients in the SA group and 3 patients in the S-1 group. The HR was 1.249 (95% CI; 0.3377–4.616; Fig. 6A). Delayed cervical lymph node metastasis developed in 14 patients in the SA group and 2 patients in the S-1 group. The HR was 2.809 (95% CI; 0.6377–12.38; Fig. 6B). In the OC, comparison of the survival time from local recurrence or delayed cervical lymph node metastasis between the treatment groups revealed that the HR for death was 4.271 (95% CI; 0.5411–32.680) in the SA group compared to the S-1 group (Fig. 5C). In contrast, in the PSMC, local recurrence developed in 4 patients in the SA group and 3 patients in the S-1 group. The HR was 1.869 (95% CI; 0.4157– 8.404; Fig. 6A). Delayed cervical lymph node metastasis developed in 5 patients in the SA group and 2 patients in the S-1 group. The HR was 3.191 (95% CI; 0.6152–16.55; Fig. 6B). In the PSMC, comparison of the survival time from local recurrence or delayed cervical lymph node metastasis between the treatment groups indicated that the HR for death was 5.691 (95% CI; 0.6637–48.80) in the SA group compared to the S-1 group (Fig. 6C).”

We have also added the following text in the Discussion section:

Page 16, lines 271–277 

“Although there was no statistically significant difference between the groups, the present data, including the results of explanatory data analyses, showed that the cumulative rates of local recurrence and delayed cervical lymph node metastasis in the S-1 group tended to be smaller than those in the SA group in both the OC and PSMC. In addition, the time from recurrence or delayed cervical lymph node metastasis to death tended to be longer in the S-1 group. As observed in a previous study (ACTS-HNC) [13], these results possibly indicate that S-1 contributes to disease control after loco-regional failure in patients with OSCC.”

We thank you again for your helpful suggestions, and hope that you will now find the revised manuscript suitable for publication.

---

## [Decision Letter · Decision Letter 1]

30 Mar 2020

Efficacy of adjuvant chemotherapy with S-1 in stage II oral squamous cell carcinoma patients: A comparative study using the propensity score matching method

PONE-D-19-27210R1

Dear Dr. Yoshida,

We are pleased to inform you that your manuscript has been judged scientifically suitable for publication and will be formally accepted for publication once it complies with all outstanding technical requirements.

With kind regards,

Jason Chia-Hsun Hsieh, M.D. Ph.D

Academic Editor

PLOS ONE

Additional Editor Comments (optional):

All of the questions were answered adequately.

Reviewers' comments:

Reviewer's Responses to Questions

**Comments to the Author**

1. If the authors have adequately addressed your comments raised in a previous round of review and you feel that this manuscript is now acceptable for publication, you may indicate that here to bypass the “Comments to the Author” section, enter your conflict of interest statement in the “Confidential to Editor” section, and submit your "Accept" recommendation.

Reviewer #2: All comments have been addressed

Reviewer #3: All comments have been addressed

2. Is the manuscript technically sound, and do the data support the conclusions?

Reviewer #2: Yes

Reviewer #3: Partly

3. Has the statistical analysis been performed appropriately and rigorously? 

Reviewer #2: Yes

Reviewer #3: Yes

4. Have the authors made all data underlying the findings in their manuscript fully available?

Reviewer #2: Yes

Reviewer #3: Yes

5. Is the manuscript presented in an intelligible fashion and written in standard English?

Reviewer #2: Yes

Reviewer #3: Yes

6. Review Comments to the Author

Reviewer #2: The manuscript was significantly improved. The authors addressed most of my concerns. This manuscript is suitable for PLOS ONE.

Reviewer #3: The author has extensively revised the manuscript.

Some additional exploratory analysis may contribute to the limitations of few samples and short following time

7. PLOS authors have the option to publish the peer review history of their article (what does this mean?). If published, this will include your full peer review and any attached files.

Reviewer #2: No

Reviewer #3: No

---

## [Editor Report · Acceptance letter]

1 Apr 2020

PONE-D-19-27210R1 

Efficacy of adjuvant chemotherapy with S-1 in stage II oral squamous cell carcinoma patients: A comparative study using the propensity score matching method 

Dear Dr. Yoshida:

I am pleased to inform you that your manuscript has been deemed suitable for publication in PLOS ONE. Congratulations! Your manuscript is now with our production department. 

With kind regards,

on behalf of

Dr. Jason Chia-Hsun Hsieh 

Academic Editor

PLOS ONE